# Therapist-supported online cognitive therapy for post-traumatic stress disorder (PTSD) in young people: protocol for an early-stage, parallel-group, randomised controlled study (OPTYC trial)

Patrick Smith [1,2] Anke Ehlers,[3] Ewan Carr [4] David Clark,[3] Tim Dalgleish,[5,6] Gordon Forbes,[4] Kimberley Goldsmith,[4] Helena Griffiths,[1] Monica Gupta,[1] Dorothy King,[1] Sarah Miles,[1,2] Dominic Plant,[1] William Yule,[1] Richard Meiser-Stedman [7]

**Correspondence to**
Dr Patrick Smith;
patrick.smith@kcl.ac.uk

## ABSTRACT

**Introduction** Post-traumatic stress disorder (PTSD) is a disabling psychiatric condition that affects a significant minority of young people exposed to traumatic events. Effective face-to-face psychological treatments for PTSD exist. However, most young people with PTSD do not receive evidence-based treatment. Remotely delivered digital interventions have potential to significantly improve treatment accessibility. Digital interventions have been successfully employed for young people with depression and anxiety, and for adults with PTSD. However, digital interventions to treat PTSD in young people have not been evaluated. The Online PTSD Treatment for Young People & Carers (OPTYC) trial will evaluate the feasibility, acceptability and initial indications of clinical efficacy of a novel internet-delivered Cognitive Therapy for treatment of PTSD in young people (iCT-PTSD-YP).

**Methods and analysis** This protocol describes a two-arm, parallel-groups, single-blind (outcome assessor), early-stage randomised controlled trial, comparing iCT-PTSD-YP with a waiting list (WL) comparator. N=34 adolescents (12–17 years old), whose primary problem is PTSD after exposure to a single traumatic event, will be recruited from 14 NHS Child and Adolescent Mental Health Services in London and southeast England, from secondary schools and primary care in the same region, or via self-referral from anywhere in the UK using the study website. Individual patient-level randomisation will allocate participants in a 1:1 ratio, randomised using minimisation according to sex and baseline symptom severity. The primary study outcomes are data on feasibility and acceptability, including recruitment, adherence, retention and adverse events (AEs). The primary clinical outcome is PTSD diagnosis 16 weeks post-randomisation. Secondary clinical outcomes include continuous measures of PTSD, anxiety and depression symptoms. Regression analyses will provide preliminary estimates of the effect of iCT-PTSD-YP on PTSD diagnosis, symptoms of PTSD, anxiety and depression relative to WL. Process-outcome evaluation will consider which mechanisms mediate recovery. Qualitative interviews with young people, families and therapists will evaluate acceptability.

**Ethics and dissemination** The study was approved by a UK Health Research Authority Research Ethics Committee (19/LO/1354). For participants aged under 16, informed consent will be provided by carers and the young person will be asked for their assent; participants aged 16 years or older can provide informed consent without their parent or caregiver's involvement. Findings will be disseminated broadly to participants, healthcare professionals, the public and other relevant groups. Study findings will be published in peer-reviewed journals.

**Trial registration number** ISRCTN16876240.

## Strengths and limitations of this study

► An early-stage trial to gather data on feasibility, acceptability and initial indications of clinical efficacy of internet-delivered Cognitive Therapy for post-traumatic stress disorder in young people (iCT-PTSD-YP).
► Young people were extensively involved in designing the phone application and website.
► CT-PTSD is theory-based and has demonstrated efficacy when delivered face-to-face and iCT-PTSD is effective in adults.
► This trial can be delivered entirely remotely.
► This early stage randomised controlled trial is not powered to detect between group effects.

## INTRODUCTION
### Background and rationale

Trauma exposure and post-traumatic stress disorder (PTSD) are both prevalent among youth under 18 years old. Between 15%–82%

of youth are exposed to traumas, and between 3%–8% of youth will develop PTSD by the age of 18 years,[1–3] representing a significant level of morbidity for health services. For affected individuals, PTSD is highly distressing, causes marked impairments in functioning and may run a chronic course for years or decades if left untreated.[4 5]

Effective treatments for PTSD exist. Recent reviews of psychological treatments for PTSD in youth find that various forms of trauma-focused cognitive behavioural therapy (TF-CBT) show consistently large effects in reducing PTSD symptoms and associated comorbidities.[6 7] Cognitive therapy for PTSD (CT-PTSD) is a form of TF-CBT developed by our group[8 9] recommended as a first-line intervention in national and international practice guidelines.[10] The treatment is theory-based, manualised and delivered over 10–12 individual sessions. Two published randomised controlled trials (RCTs)[11 12] find that CT-PTSD is acceptable to young people (8–18 years old), and efficacious.[13]

However, most young people under 18 years old with PTSD do not receive an effective, evidence-based treatment. The gap between community prevalence of psychiatric disorders and treatment provision for young people is well-known and longstanding.[14] In a recent population based British study, only 40% of young people with PTSD sought help from general practitioners (GPs) or mental health practitioners and only 20% had accessed specialist mental health services in the past year.[15] Limited access to treatment may be due to multiple interacting factors including under-capacity and long waiting times for assessment and treatment in specialist Child and Adolescent Mental Health Services,[16] and the burden and inconvenience to young people and families in attending face-to-face appointments in a clinic.

Remote delivery of psychological therapy via the internet has enormous potential to address some of these barriers, and to increase accessibility of treatment.[17] Young people have enthusiastically endorsed the potential for digital health interventions.[18] For disorders other than PTSD, digital health interventions are known to be acceptable to young people and clinically helpful. For example, computerised cognitive behavioural therapy (C-CBT) for depression demonstrates clear clinical benefit for young people[19 20] and is now recommended by National Institute for Health and Care Excellence.[21] Lessons have been learnt about the development of digital mental health interventions including the need for co-design with young people,[22] and the active engagement of young people in therapy facilitated by continued therapist support during treatment.[19]

Development of remotely delivered therapy for treatment of PTSD in young people lags behind that for other disorders. Jaycox and colleagues[23] report encouraging preliminary outcomes for a self-help web-based tool to augment and enhance usual school support services for trauma-exposed youth (7th – 12th grade, mean age 15 years). Kassam-Adams and colleagues[24] showed that a digital intervention for preventing PTSD symptoms in

injured children (8–12 years old) was feasible and clinically promising. Ruggiero and colleagues[25] found that use of a web-based psychoeducation intervention for disaster-affected adolescents (mean age 14.5 years) was associated with improvements in PTSD symptoms. However, to our knowledge, no studies have yet reported on the development or evaluation of internet-delivered TF-CBT for treatment of PTSD in children and young people. This is surprising because face-to-face TF-CBT is well established as an effective treatment for PTSD in youth, and work with adults shows that PTSD is a disorder which is treatable via the internet.[26]

In this project we aim to address this clear gap. We have co-designed with adolescents an internet version of CT-PTSD, to be delivered via smartphone application and website, with remote therapist support. Our longer-term intention is to determine whether this approach will help to reduce the treatment gap for young people with PTSD by making an efficacious therapy more widely available. Our aim in the current early-stage trial is to gather preliminary data on feasibility, acceptability and initial signal of clinical effects of internet-delivered cognitive therapy for treatment of PTSD in young people (iCT-PTSD-YP), relative to a waiting list (WL) condition. Data gathered in the current trial will be used to inform the design and size of a future scaled-up trial. All items from the WHO Trial Registration data set are detailed in online supplemental appendix 1.

## Objectives

The primary objective is to provide data on feasibility, acceptability, compliance, retention and delivery of iCT-PTSD-YP. The secondary objective is to provide initial estimates of the effect of iCT-PTSD-YP on symptoms of PTSD, anxiety and depression relative to a WL condition.

## METHODS AND ANALYSIS
### Trial design

This study is a two-arm, parallel groups, single-blind (outcome assessor), early stage RCT, comparing iCT-PTSD-YP with a WL comparator. Individual patient-level randomisation will allocate participants in a 1:1 ratio, randomised using minimisation according to sex and baseline symptom severity.

### Patient and public involvement

Members of the NIHR Maudsley Biomedical Research Centre (BRC) Young Person's Mental Health Advisory Group (YPMHAG; 16–25 year-olds with lived experience of using mental health services) were consulted before grant submission: they provided verbal and written feedback on the research ideas. Young people (N=33, aged 12–17 years old) were consulted at an early stage about the design of the application via a series of four focus groups held in four different schools. Young people receiving face-to-face CT-PTSD provided feedback on initial prototypes of the application. A young person

with lived experience of using mental health services is a member of the Trial Steering Committee (TSC). We will consult the YPMHAG and the TSC about our dissemination strategy.

## Study setting

The trial will be carried out in the UK. Trial randomisation will be carried out by King's College London Clinical Trials Unit (CTU). Trial therapists will be based at King's College London and the University of East Anglia. Referrals will be sought from 14 NHS Child and Adolescent Mental Health Services (CAMHS) in London and southeast England, all of which are registered as study sites. Referrals will also be sought from secondary schools and primary care in the same region. We will offer to carry out screening surveys in schools to identify potentially eligible young people (12–17 years old). Self-referral from anywhere in the UK is also possible via the study website.

## Eligibility criteria

Young people are eligible to be included if: they are aged 12–17 years old; their main presenting problem is PTSD and there is a not a comorbid problem that would preclude treatment of PTSD; PTSD symptoms relate to a single trauma; they speak English to a level that allows therapy without the need for an interpreter, and they read English to a level that allows independent use of iCT; they have access to a smartphone and a larger device (laptop, desktop computer, tablet) with internet access, and they have access to a safe and confidential space in which to engage in iCT. Young people are excluded if they have: brain damage; intellectual disability; pervasive developmental disorder or neurodevelopmental disorder, as assessed by clinical interview with parents/carers; other psychiatric diagnosis that requires treatment before PTSD, determined by clinical interview and questionnaires; moderate-to-high risk to self; ongoing trauma-related threat; have started treatment with psychotropic medication, or changed medication, within the last 2 months; or are currently receiving another psychological treatment, as assessed in clinical interview; or previously received TF-CBT in relation to the same traumatic event that they are currently seeking treatment for.

Parents or carers are eligible to be included if they: are the parent or carer of a young person who meets all of the inclusion criteria and none of the exclusion criteria above; speak English to a level that allows participation in therapy without the need for an interpreter, and read English to a level that allows independent use of iCT; and have access to a smartphone and/or larger device with internet access.

## Interventions
### iCT-PTSD-YP

iCT-PTSD-YP comprises therapist-supported online delivery of all components from our published manual of face-to-face CT-PTSD for young people.[27] Treatment aims to change

problematic appraisals, update trauma memories and change unhelpful coping responses. Treatment components are delivered in modules. There are 10 core modules for all young people (Psychoeducation about PTSD, Reclaiming life, Understanding PTSD, Developing a trauma narrative, Identifying hotspots, Updating the narrative, Working with triggers, Overcoming sense of danger, Visiting the site virtually and/or in person, Developing a blueprint) that are released to the young person sequentially by the therapist, and 11 optional modules which are released according to individual need (Relaxation, Sleep, Working with images, Working with physical difference, Anger, Grief, Shame, Guilt, Self-criticism, Rumination and Panic). Modules were co-designed with input from young people and built on the content of the modules developed for iCT-PTSD for adults.[28 29] Modules are interactive (prompting for user action to progress through the application and requesting user text input and questionnaire responses) and include text, illustrations, audio case examples, animations and videos. Modules are intended for independent self-study by young people. Therapists can log onto the site to view young people's progress including their text input and questionnaire responses. Young people and therapists can message each other via the application. Parents and carers are provided a separate log on to the carer version of the application. The carer version comprises eight modules, and the emphasis is on providing information to carers about therapy, including advice about how carers can help in young people's recovery. Carers do not have access to any information that their child inputs to the application. Modules are delivered via a progressive web application on a smartphone or computer, hosted on a secure server. The application is not publicly available currently. For trial participants, an individual account requiring two-factor authentication log-in is created for the young person and their carer.

Therapists will be clinical psychologists or CBT therapists who have received training in face-to-face CT-PTSD, and in use of the iCT-PTSD-YP application. Therapists will have contact with young people and carers via phone or videoconferencing at least once a week for the duration of therapy. Therapists release modules according to the young person's individual formulation, remind and encourage young people to log on to the application and provide support in using the application and implementing the treatment components. Weekly clinical supervision will be provided by a consultant clinical psychologist from the trial team.

Therapy is delivered over 12 weeks. Post-treatment assessment is carried out 1 month after the end of treatment (ie, at 16 weeks after randomisation).

## WL

Young people will be placed on a WL and re-assessed 16 weeks after randomisation. Young people who require treatment at the end of the waiting period will be offered immediate iCT-PTSD-YP. WL control arms are commonly used in PTSD treatment trials[6] because natural recovery from PTSD can be substantial.[30] Use of a WL condition ensures that the effect of treatment is not overestimated,

and shows whether treatment is impeding the rate of natural recovery.

## Withdrawals

Participants will be withdrawn from treatment if: a current illness prevents further treatment; there is a change in the participant's condition or circumstances that in the clinician's opinion justifies the discontinuation of treatment; or the participant withdraws consent for treatment. Participants who discontinue treatment for the above reasons will be invited to provide follow-up data and will remain in the trial for the purposes of data analysis. If the participant no longer wishes to be followed up to provide research data, the participant will be withdrawn entirely

from the trial. The different types of withdrawal will be captured and reported.

## Outcomes

The schedule for assessments is presented in table 1.

The primary outcomes for the study are data on feasibility, adherence and acceptability, which will be reported using the metrics specified below.

### Feasibility outcomes

We will report: (1) the number of young people referred to the trial in total and according to referral route; (2) the number of young people screened in schools, and the proportion of those who proceed to a phone call with the

**Table 1** Study schedule

| Measure | Study period | | | | | |
| --- | --- | --- | --- | --- | --- | --- |
| | Screen 0–1 week | Pre 0 weeks | Weekly (*iCT only*) | Mid 0+6 weeks | Post 0+16 weeks | Follow-up 0+38 weeks (*iCT only*) |
| Enrolment | | | | | | |
| Eligibility screen | x | | | | | |
| Provide study information | x | | | | | |
| Gain informed consent | | x | | | | |
| Online assessment | | | | | | |
| Dawba | | x | | | | |
| Interview | | | | | | |
| Demographic interview | | | | | | |
| CAPS-CA-5 | | x | | | x | |
| CGAS | | x | | | x | |
| *Adolescent questionnaires* | | | | | | |
| CPSS-5 | | x | | | x | x |
| CRIES-8 | | x | x | x | x | x |
| RCADS-C | | x | | | x | x |
| CPTCI | | x | | x | x | x |
| TMQQ | | x | | x | x | x |
| Rumination items | | x | | x | x | x |
| CHU-9D | | x | | | x | x |
| Adverse events | | | | x | x | x |
| Carer questionnaires | | | | | | |
| SDQ-P | | x | | | x | x |
| RCADS-P | | x | | | x | x |
| CA-SUS | | x | | | x | x |
| Adverse events | | | | x | x | x |
| Qualitative interviews | | | | | | |
| Adolescents | | | | | x | |
| Carers | | | | | x | |
| Therapists | | | | | x | |

CAPS-CA-5, Clinician Administered PTSD Scale for DSM-5: Child and Adolescent version; CA-SUS, Child and Adolescent Service Use Schedule; CGAS, Children's Global Assessment Scale; CHU-9D, Child Health Utility Index 9D; CPSS-5, Child PTSD Symptom Scale for DSM-5; CPTCI, Child Post Traumatic Cognitions Inventory; CRIES-8, Children's Revised Impact of Event Scale, 8-item version; DSM-5, Diagnostic and Statistical Manual of Mental Disorders; iCT, internet-delivered cognitive therapy; PTSD, post-traumatic stress disorder; RCADS-C, Revised Children's Anxiety and Depression Scale—child version; RCADS-P, RCADS—parent version; SDQ-P, Strength and Difficulties Questionnaire—parent version; TMQQ, Trauma Memory Quality Questionnaire.

family; (3) the number and proportion of young people in schools scoring above cut-off on a validated screening questionnaire (CRIES-8, see below) relative to the number of young people screened in schools; (4) the number and proportion of young people in schools who score above cut-off on the screening questionnaire but decline further participation with the trial relative to those scoring above cut-off); (5) the number and proportion of young people in schools who score above cut-off on the screening and consent to further assessment but are deemed ineligible at baseline assessment relative to those deemed eligible at baseline assessment; (6) the number of assessment appointments offered to participants; (7) the number and proportion of assessment appointments attended by participants, relative to the number of appointments offered, reported by referral source; (8) reasons for not attending assessment appointments, reported by referral source; (9) the number and proportion of young people who at baseline assessment consent to participate in the trial, relative to the number who attend assessment, with reasons for not consenting if known; (10) the number and proportion of young people eligible for the trial after baseline assessment, relative to the number of baseline assessments completed; (11) the number and proportion of young people who are randomised, and the proportion of consented young people who are randomised relative to the number who consented; (12) reasons for withdrawing from the trial if known; and (13) the number retained in study at 16 weeks (post-treatment) and at 38 weeks (follow-up), and the proportions of those who start treatment who are retained.

### Adherence metrics

For participants allocated to iCT-PTSD-YP, we will report: (1) the number of times logged into the programme per week and in total; (2) time spent logged in per week and in total; (3) the number of modules completed in total and according to device used; (4) the number of therapist phone calls attended per week and in total, and the number of missed phone appointments; (5) time spent on phone calls per week and in total; (6) the number of messages to/from therapist per week and in total; (7) the number and proportion of young people who start treatment; (8) the number of weeks of therapy completed and (9) reasons for dropping out of treatment if known.

### Acceptability outcomes

We will carry out qualitative interviews with young people, carers and therapists to gauge acceptability of iCT-PTSD-YP, and we will summarise interview data using content analysis. We will aim for these interviews to be representative of individuals involved in the feasibility trial (young people, carers, therapists), including young people who left the study or failed to adhere to the course of treatment, to provide a full range of views. We will interview trial participants in both arms about the acceptability of the research procedures including the assessment measures and their views on randomisation.

### Primary clinical outcome

Presence of PTSD according to the Diagnostic and Statistical Manual of Mental Disorders (DSM-5) at 16 weeks post-randomisation, ascertained using the Clinician Administered PTSD Scale for DSM-5: Child and Adolescent version (CAPS-CA-5[30]), administered by trained reliable raters, blind to treatment allocation.

### Secondary clinical outcomes

Child-reported outcomes at 16 weeks post randomisation: PTSD symptom severity (continuous score) on the CAPS-CA-5;[31] PTSD symptom severity on the Child PTSD Symptom Scale for DSM-5 (CPSS-5[32]); PTSD symptom severity on the Children's Revised Impact of Event Scale, 8-item version (CRIES-8[33 34]); and symptoms of depression and anxiety on the 25-item Revised Children's Anxiety and Depression Scale (RCADS).[35] Carer reported outcomes at 16 weeks post randomisation: Revised Children's Anxiety and Depression Scale—parent version (RCADS-P[35]); and Strength and Difficulties Questionnaire—parent version (SDQ-P).[36] At 38-week follow-up for participants in the iCT-PTSD-YP only, all secondary clinical outcomes apart from the CAPS-CA-5 will be repeated.

### Process measures

The cognitive model[8] on which treatment is based specifies a number of mechanisms of therapeutic change. We will test mediation via changes in appraisals, memory quality and ruminative thinking from baseline to mid-treatment (6 weeks post randomisation) using: the Child Post Traumatic Cognitions Inventory (CPTCI[37]); the Trauma Memory Quality Questionnaire (TMQQ[38]); and the Trauma Related Rumination Questionnaire items.[39]

### Health economic outcomes

We will collect economic data on health utilities and resource use using the Child Health Utility Index 9D (CHU-9D)[40] and the Child and Adolescent Service Use Schedule,[41] administered at baseline and 16 weeks post-randomisation.

### Participant timeline

All participants will be assessed three times during the study: pre-treatment (week 0), mid-treatment (week 6 post-randomisation) and post-treatment (week 16 post-randomisation). Participants in iCT-PTSD-YP will complete a brief weekly measure of PTSD symptoms (CRIES-8) and mood (Likert scale) on the application, and a follow-up assessment (week 38 post randomisation). The first participant was randomised on 24 August 2020, and the last participant was randomised on 20 October 2021. The trial is currently closed to new recruitment.

### Sample size

We will recruit 17 participants per arm. In our previous RCTs of face-to-face CT-PTSD[11 12] in young people, we had 4% drop-out, but we have conservatively allowed for approximately 20% drop-out, to give at least n=14 at post-treatment in each arm. An early-stage trial of this size

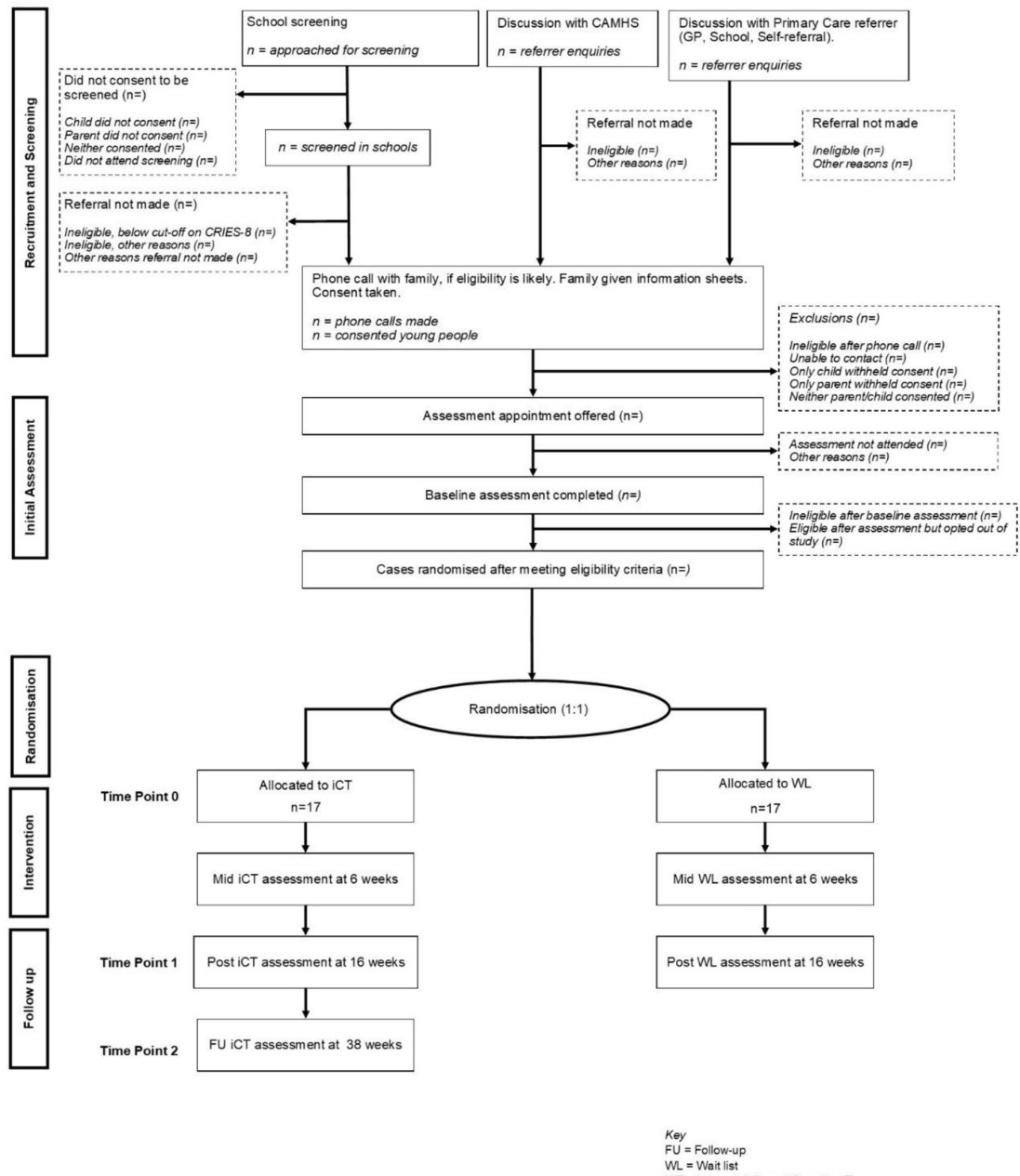

**Figure 1** Study flowchart. CAMHS, NHS Child and Adolescent Mental Health Services; CRIES-8, Children's Revised Impact of Event Scale, 8-item version; GP, general practitioner.

will be sufficient to gather meaningful feasibility data on acceptability, compliance, retention and delivery. Power calculations are not typically used to determine sample size for feasibility studies. Therefore, we acknowledge an insufficient sample size to allow definitive between-group comparisons in this early stage RCT.[42 43]

### Recruitment

Participants will be recruited via three routes (see figure 1): (1) from school screening; (2) from NHS CAMHS teams; and (3) from primary care (GP or school referral) or self-referral. For all referral routes, consent will be sought before assessment, and eligibility will be determined by the clinical assessment.

### Allocation

Once a participant is confirmed as eligible and consenting to the study, they will be registered in the main participant database (held using the IBM-SPSS program). Participants will be randomised to receive iCT-PTSD-YP or WL at a 1:1 ratio. Randomisation will be carried out by the King's College London CTU via a web-based service utilising minimisation with a random component. Minimisation factors will be sex and baseline PTSD symptom severity assessed by the CPSS (low:<51 and high:≥51). These factors were chosen in order to balance factors that may affect treatment response across the two arms. Other factors (such as age

and trauma type) were not included due to the modest trial size.

## Blinding

All assessors of the primary and secondary clinical outcomes at follow-up at 16 weeks will be blind to trial arm allocation. Blind outcome assessors will be independent research assistants or clinical psychologists who are not part of the trial team. Assessors will be trained to standard on the CAPS-CA-5 interview, and inter-rater reliability will be assessed for 20 randomly selected interviews. The senior trial statistician (KG) will also be blind with all other members of the study team unblind to trial arm allocation. Unblinding of the senior trial statistician and the analysis of outcomes by intervention arm will occur after the initial draft of the statistical analysis report is generated.

## Data collection methods

For the primary clinical outcome, the CAPS-CA clinical interview is completed on the phone or via videoconference, with symptom level responses marked on the interview form and then entered into the trial database. For secondary clinical outcomes, questionnaires are completed online via a secure commercial system (Qualtrics) with responses downloaded to an electronic database and re-entered into the trial database. Feasibility outcomes are recorded by the study research assistant in the trial database. Adherence metrics are either recorded by the trial therapist in the study database or automatically captured by the application and downloaded to standard database software.

## Data management

Participant information will be kept confidential and managed in accordance with the Data Protection Act, General Data Protection Regulation (GDPR) policies, NHS Caldicott Guardian, The UK Policy Framework for Health and Social Care Research and Research Ethics Committee Approval. Personally identifiable data will be collected from participants including name and contact details. This information will be stored securely and separately from all other study-generated data, which will be anonymised. Each participant will be given a unique Participant Identification Number (PIN). All feasibility and clinical outcomes for the RCT will be stored in SPSS databases against the participant PIN. These databases will be stored on a secure King's College London network drive, accessible to the study team only. Databases will be stored in a version control system, such that changes made over time can be examined and recovered. All databases will be registered in the King's Data Protection Register.

## Statistical methods

A comprehensive statistical analysis plan (SAP) will be developed and agreed with the TSC before any analysis is carried out. The SAP will describe statistical procedures in detail. Quantitative analyses will employ up-to-date versions of statistical software (eg, Stata or R).

## Analysis of feasibility outcomes and adherence metrics

The feasibility outcomes and adherence metrics will be summarised with appropriate summary statistics (eg, means and SD/medians and IQRs for continuous outcomes; frequencies and proportions for count outcomes). Where appropriate some feasibility outcomes will either be reported only for the iCT-PTSD-YP arm or will be reported separately by arm.

## Clinical outcomes

As this is an early-stage trial designed to gather data on feasibility outcomes, it is not powered to detect between-arm differences: where between-arm differences are presented, they will be treated as exploratory and not treated as inferential. Data completeness will be summarised for clinical outcomes. All comparative analyses will primarily be conducted under the intention-to-treat (ITT) principle—all participants with a completed outcome will be included in the analysis and analysed according to the arm they were randomised to. Where deviations from ITT occur, this will be reported. We will carry out per-protocol analyses in addition to ITT, but these analyses will be treated as secondary to the ITT analysis. There will be no interim or subgroup analyses.

The primary and secondary clinical outcomes will be summarised with appropriate summary statistics by trial arm at each time point (primary, frequencies and proportions; secondary, means and SD). For each outcome we will estimate the treatment effect at 16 weeks, with the appropriate 95% CI. The iCT-PTSD-YP versus WL OR for remission from PTSD caseness at 16 weeks post-randomisation will be assessed using logistic regression with trial arm and the minimisation variables as covariates. The iCT-PTSD-YP versus WL mean differences in secondary clinical outcomes at 16 weeks post-randomisation will be estimated using linear regression, with trial arm, baseline outcome score and minimisation variables as covariates.

We will carry out per-protocol analyses for the primary outcome, and the CPSS-5 and CRIES-8 secondary outcomes at 16 weeks. These will be treated as secondary to the ITT analysis. The per-protocol analyses will be conducted in two populations. The first will consist of all participants with recorded outcome data who complete the minimum therapy needed to achieve clinical benefit (defined as completing at least the first six core modules (Psychoeducation about PTSD, Reclaiming life, Understanding PTSD, Developing a trauma narrative, Identifying hotspots, Updating the narrative)). The second per-protocol population will consist of all participants from the first per-protocol population who have additionally completed the core module, 'Working with triggers'.

## Process outcomes

An exploratory mediation analysis will be carried out to assess the indirect effect of treatment allocation on the primary clinical endpoint via the CPTCI, the TMQQ and items relating to rumination, measured at 6 weeks post-randomisation. The total, direct and indirect effects of

treatment allocation on 16-week PTSD caseness will be estimated using the Stata paramed command[44 45] to properly calculate effects for a binary outcome, along with associated 95% CIs. CIs for the indirect effect will be estimated using the percentile bootstrap.[46]

## Health economics

To gauge the feasibility of collecting health economic data, data completeness will be summarised by presenting the number and proportion of complete and missing values at each time point. Efficacy will be measured using the CHU-9D measure of health-related quality of life. Data on iCT-PTSD-YP, contact time and indirect time for the intervention will be collected directly from clinicians and service records. Service use estimates will be combined with standard UK sources for unit costs to estimate total costs. The cost of iCT-PTSD-YP will be directly calculated. These data will allow us to index service use and permit preliminary estimates of the potential cost-effectiveness of iCT-PTSD-YP.

## Qualitative analysis

We will carry out qualitative interviews at the end of each participant's iCT-PTSD-YP. If participants drop out of treatment early, we will endeavour to interview them. Semi-structured interviews using a topic guide will be carried out by a member of the study team who was not involved in treatment. The views and experiences of patients, parents or carers and trial clinicians will be sought in order to gain a multiperspective view of acceptability. Content analysis will be used to explore both commonalities and variations within and between these respondents. We will interview trial participants in both arms about the acceptability of the research procedures including the assessment measures and their views on randomisation. We will invite all participants to take part in qualitative interviews, until data saturation is reached.

## Data monitoring

Project oversight will be provided by a monthly Project Management Group (PMG) attended by all co-investigators. Trial oversight will be provided by a 6-monthly TSC. The TSC will review the protocol, agree the SAP and safeguard the interests of trial participants. The TSC will provide advice to the chief investigator and sponsor. A separate Data Monitoring Committee (DMC) will not be convened. The TSC will monitor AEs and adverse reactions and will convene an emergency DMC if needed.

## AEs

AEs are defined as any untoward occurrence in a trial participant, including events that are not necessarily caused by or related to trial procedures. Serious AEs are defined as AEs that result in death, are life-threatening, require hospitalisation or prolong existing hospitalisation or result in persistent or significant disability or incapacity. Some AEs are expected in this study, and will be reported to the TSC, for example: self-harm not requiring medical attention, increase in suicidal ideation, worsening of PTSD symptoms (defined as 7-point increase in CRIES-8). Serious AEs will be reported to the Chair of the TSC, the Research Ethics Committee (REC) and the sponsor. AEs will be assessed at each assessment time point. Risk monitoring including AE monitoring will be carried out during clinical contact for those allocated to iCT-PTSD-YP. AEs will be monitored and recorded from randomisation to final follow-up.

# ETHICS AND DISSEMINATION
## Ethical approval

The study was approved by a UK Health Research Authority (HRA) Research Ethics Committee (REC; 19/LO/1354). The study is sponsored by King's College London.

## Protocol amendments

We were initially funded to run a three-arm feasibility RCT comparing iCT-PTSD-YP with face-to-face CT-PTSD and WL. The COVID-19 pandemic national lockdown was implemented before we started to recruit to the planned three-arm trial. Restrictions in CAMHS services due to lockdown meant that we could not offer face-to-face CT-PTSD. Therefore, after consultation with the funder and the TSC we changed the design to the current two-arm trial and received HRA and REC approval to proceed. This change was made before recruitment started, and before registration on ISRCTN.

Further protocol amendments will require approval from the REC, and where relevant will be passed on to the trial register.

## Consent and assent

For participants aged under 16, informed consent will be provided by carers and the young person will be asked for their assent. Participants aged 16 years or older can provide informed consent without their parent or caregiver's involvement. Please see online supplemental file for copies of consent and assent forms.

## Confidentiality

Information with regards to participants will be kept confidential. The treating clinician and research team involved in day-to-day trial management will have access to personally identifiable data so that they can maintain contact with participants throughout the study. Participants will be assigned a study ID. All outcome data will be stored against this study ID so that data are anonymised.

## Access to data

All investigators will have access to the final trial data set. Our intentions are to maximise the availability and sharing of our data for the benefit of the wider research community, while providing for its long-term preservation and making due allowance for the potential commercial value of findings. The PMG will make the decision on whether to supply research data to a potential new researcher. Independent oversight of data access and sharing will be

provided by the TSC. Data released to the wider community after publication will be fully anonymised.

## Dissemination policy

There are no publication restrictions and findings will be disseminated broadly to participants, healthcare professionals, the public and other relevant groups. The study findings will be published in peer-reviewed journals. The full trial protocol is available from PS.

## DISCUSSION

PTSD in children and adolescents is a significant public health burden. Highly efficacious treatments exist but are not widely accessible. Remotely delivered iCT-PTSD has potential to facilitate a step change in improving accessibility of an evidence-based therapy for youth. The data gathered in the current trial will inform the design and size of a future scaled up trial to evaluate remotely delivered iCT-PTSD-YP.

**Author affiliations**
[1]Department of Psychology, Institute of Psychiatry, Psychology and Neuroscience, King's College London, London, UK
[2]South London & Maudsley NHS Foundation Trust, London, UK
[3]Department of Experimental Psychology, University of Oxford, Oxford, UK
[4]Department of Biostatistics and Health Informatics, Institute of Psychiatry, Psychology and Neuroscience, King's College London, London, UK
[5]Medical Research Council Cognition and Brain Sciences Unit, University of Cambridge, Cambridge, UK
[6]Cambridgeshire and Peterborough NHS Foundation Trust, Cambridge, UK
[7]Department of Clinical Psychology and Psychological Therapies, University of East Anglia, Norwich, UK

**Acknowledgements** We are very grateful to the young people who helped to shape the project and made key contributions to the design of the intervention, and to the young people and carers who participate in the trial. We are very grateful to the Trial Steering Committee (Cathy Creswell, Andrew Brand, Rachel Calam and Paul Stallard) for their advice and support. EC, GF and KG's contributions represent independent research part funded by the NIHR Biomedical Research Centre (South London and Maudsley NHS Foundation Trust and King's College London). KG receives funding from the NIHR Applied Research Collaboration South London (King's College Hospital NHS Foundation Trust). The views expressed are those of the authors and not necessarily those of the NHS, the NIHR, the Department of Health and Social Care.

**Contributors** PS, DC, TD, AE, KG, RM-S and WY designed the trial. PS, DC, TD, AE, HG, MG, DK, RM-S, SM, DP and WY contributed to application development and delivery. PS, DC, EC, TD, AE, GF, KG, HG, DK, RM-S, SM and WY oversaw recruitment and data collection. PS drafted the protocol. All authors read and approved the final manuscript. All authors have agreed both to be personally accountable for their own contributions and to ensure that questions related to the accuracy or integrity of any part of the work, even ones in which the author was not personally involved, are appropriately investigated, resolved and the resolution documented in the literature.

**Funding** OPTYC is funded by the UK Medical Research Council's Developmental Pathway Funding Scheme (Grant reference: MR/P017355/1). TD was funded by the UK Medical Research Council (Grant Reference: SUAG/043 G101400). KG receives funding from the NIHR Applied Research Collaboration South London (King's College Hospital NHS Foundation Trust).

**Competing interests** Some authors (DC, TD, AE, RM-S, DP and PS) provide training in the delivery of CT-PTSD, for which they may sometimes receive payment. PS, DC and WY are coauthors on a published treatment manual of CT-PTSD for children and young people and receive royalties from sales.

**Patient consent for publication** Not applicable.

**Provenance and peer review** Not commissioned; externally peer reviewed.

**ORCID iDs**
Patrick Smith http://orcid.org/0000-0002-0743-7972
Ewan Carr http://orcid.org/0000-0002-1146-4922
Richard Meiser-Stedman http://orcid.org/0000-0002-0262-623X

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
