## [Reviewer comments · BMJ Open]

ARTICLE DETAILS

TITLE (PROVISIONAL)	Therapist-supported online cognitive therapy for post-traumatic stress disorder (PTSD) in young people: protocol for an early-stage, parallel-group, randomised controlled study (OPTYC trial)
AUTHORS	Smith, Patrick; Ehlers, Anke; Carr, Ewan; Clark, David; Dalgleish, Tim; Forbes, Gordon; Goldsmith, Kimberley; Griffiths, Helena; Gupta, Monica; King, Dorothy; Miles, Sarah; Plant, Dominic; Yule, William; Meiser-Stedman, Richard

VERSION 1 – REVIEW

REVIEWER	Rossouw, J Stellenbosch University Faculty of Medicine and Health Sciences, Psychiatry
REVIEW RETURNED	29-Aug-2021

GENERAL COMMENTS	Thank you for the opportunity to review this very well written and comprehensive study protocol. The need for more easily accessible treatment to adolescents and children requires the attention this study is giving to the treatment of PTSD. I have little to comment on regarding the introduction or study design and congratulate the authors on the detail provided. The authors are required to state the starting date of the trial in the main text of the article and the estimated completion time of the research. In the allocation section it could be helpful to discuss the choice of PTSD severity and sex as pre-specified prognostic minimisation factors and perhaps why other factors were excluded, such as age, type of trauma as examples were not considered. As someone with a qualification in clinical psychology and information technology I am always excited with the use of technology to make psychological treatment more accessible. I hope the authors will also publish on the design process of the app. I think it would also be important to gauge users experience of the app (not just on the treatment) and obtain their feedback on possible improvements in the user interface during the qualitative interviews. I think the authors can in this way improve UI/UX design for the main study. It might be useful to involve a UX designer, if you have not yet done so to help with this going forward. All the best with this important work.
---

REVIEWER	Biliunaite, Ieva Linköping University, Department of Behavioural Sciences and Learning
REVIEW RETURNED	22-Sep-2021

GENERAL COMMENTS

Thank you for the opportunity to review your manuscript. It is overall a well written article covering an interesting topic. Please find my comments and suggestions listed below.

Introduction:

- The first sentence is the same as in the abstract. I suggest rephrasing it in either one or the other.
- Please provide with references in support of the last two statements on the page 9, lines 19 and 21 (the active engagement and the need for continued therapist support namely).
- Several terms, such as youth/adolescence/ children are used. Could you please clarify in text when/if your examples are in relation to the target group of the 12-17 years old.

Methods:

- Patient and Public involvement: 'Young people were consulted at an early stage about the design of the App via a series of focus groups held in schools' – could you please provide with more details, such as how many focus groups were conducted and how many of the young people attended? Do you refer to young people as the above mentioned 16–25-year-olds? Alternatively, if this has been documented elsewhere, please provide with a reference.
- Study setting: it is not clear what age group is referred to by young people. I see that in the section below the ages are defined, but I think it would be much clearer for the reader if that is done before and the terminology is kept constant throughout the protocol.
- Eligibility criteria: I was wondering, how was the 'safe and confidential space' defined? And how was this evaluated?
- Intervention modules: how are the participants assigned the modules? Can they access them all at once, or is it assigned one by one on, for example, weekly basis? I understand there are 10 core modules, 11 optional and a total of 12 weeks for receiving the treatment – are participants advised about how much they should engage with it?
- What is the purpose of the therapist contact? Is it so that participants can access questions and receive feedback? Or more for encouragement or reminders to engage with the app? Or both? Could you please clarify.
- WL group – as I understand, participants in the WL group will not necessarily be allowed to engage with the intervention (unless they still need treatment after the waiting period). Is there another reason other than stated not to allow these participants engage with the intervention once the intervention group is finished? Natural recovery argument somehow makes it sound as it is less important (to develop and evaluate such interventions) and it should certainly not be the case.
- There is a break in the page on the p15of57 – measure and study period come in the middle of the text – not sure if this was in the original file or occurred when uploading.
- Qualitative interviews: could you please provide with few more details, such as the type of interviews (structured/unstructured/semi-structured) and who will be conducting the interviews.

VERSION 1 – AUTHOR RESPONSE

Reviewer: 1

Dr. J Rossouw, Stellenbosch University Faculty of Medicine and Health Sciences, Centre for CBT
Comments to the Author:

Thank you for the opportunity to review this very well written and comprehensive study protocol. The need for more easily accessible treatment to adolescents and children requires the attention this study is giving to the treatment of PTSD. I have little to comment on regarding the introduction or study design and congratulate the authors on the detail provided.

Thank you for the encouraging comments.

The authors are required to state the starting date of the trial in the main text of the article and the estimated completion time of the research.

The date of the first and last recruited participant has been added to “Participant timeline” on page 17. In the allocation section it could be helpful to discuss the choice of PTSD severity and sex as pre-specified prognostic minimisation factors and perhaps why other factors were excluded, such as age, type of trauma as examples were not considered.

This information has been added to the “Allocation” section on page 18.

As someone with a qualification in clinical psychology and information technology I am always excited with the use of technology to make psychological treatment more accessible. I hope the authors will also publish on the design process of the app. I think it would also be important to gauge users experience of the app (not just on the treatment) and obtain their feedback on possible improvements in the user interface during the qualitative interviews. I think the authors can in this way improve UI/UX design for the main study. It might be useful to involve a UX designer, if you have not yet done so to help with this going forward.

Thank you very much for the suggestion, which we will follow through.

All the best with this important work.

Thank you.

Reviewer: 2

Ms. Ieva Biliunaite, Linköping University Comments to the Author:

Thank you for the opportunity to review your manuscript. It is overall a well written article covering an interesting topic. Please find my comments and suggestions listed below.

Thank you for the positive comments.

Introduction:

-The first sentence is the same as in the abstract. I suggest rephrasing it in either one or the other.

The first sentence has been deleted from the Introduction (page 6 para 1).

-Please provide with references in support of the last two statements on the page 9, lines 19 and 21 (the active engagement and the need for continued therapist support namely).

This has been re-phrased, and a reference (Hollis et al 2017) added (page 7 para 1)

-Several terms, such as youth/adolescence/ children are used. Could you please clarify in text when/if your examples are in relation to the target group of the 12-17 years old.

The first time that “youth” is used has been changed to “youth under 18 years old” and thereafter “youth”. (page 6 para 1).

We have clarified the ages of “young people” (i.e., 8 – 18 years old) in the two published trials cited (page 6 para 2).

We have added “under 18 years old” to clarify the ages of young people referred to in the studies cited on page 6 para 3.

We have added the age range and/or mean age of the participants in the three cited studies (page 7 para 2).

Methods:

-Patient and Public involvement: 'Young people were consulted at an early stage about the design of the App via a series of focus groups held in schools' – could you please provide with more details, such as how many focus groups were conducted and how many of the young people attended? Do you refer to young people as the above mentioned 16–25-year-olds? Alternatively, if this has been documented elsewhere, please provide with a reference.

The number of focus groups and number and ages of young people has been added to the PPI section (page 9, para 1).

-Study setting: it is not clear what age group is referred to by young people. I see that in the section below the ages are defined, but I think it would be much clearer for the reader if that is done before and the terminology is kept constant throughout the protocol.

The age information has been added to this section (page 9 para 2)

-Eligibility criteria: I was wondering, how was the 'safe and confidential space' defined? And how was this evaluated?

We asked young people and their carers whether they had access to a space that was private and safe, for example their own bedroom, from which to access the App and speak to their therapist.

-Intervention modules: how are the participants assigned the modules? Can they access them all at once, or is it assigned one by one on, for example, weekly basis? I understand there are 10 core modules, 11 optional and a total of 12 weeks for receiving the treatment – are participants advised about how much they should engage with it?

Apologies that this was not clear. We have added that the core modules are released sequentially by the therapist, and that the optional modules are released according to individual need (page 11, para 1).

-What is the purpose of the therapist contact? Is it so that participants can access questions and receive feedback? Or more for encouragement or reminders to engage with the app? Or both? Could you please clarify.

Apologies that this was not clear. We have added that the purpose of the therapist contact is to release modules according to the young person's individual formulation, to remind and encourage young people to log on to the App, and to provide support in using the App and implementing the treatment components (page 11, para 2).

-WL group – as I understand, participants in the WL group will not necessarily be allowed to engage with the intervention (unless they still need treatment after the waiting period). Is there another reason other than stated not to allow these participants engage with the intervention once the intervention group is finished? Natural recovery argument somehow makes it sound as it is less important (to develop and evaluate such interventions) and it should certainly not be the case.

All young people who require treatment after the waiting period will be offered the intervention. The purpose of the WL arm is to quantify the effect of natural recovery.

-There is a break in the page on the p15of57 – measure and study period come in the middle of the text – not sure if this was in the original file or occurred when uploading.

I will double check formatting of tables when re-uploading files, thank you for pointing this out.

-Qualitative interviews: could you please provide with few more details, such as the type of interviews (structured/unstructured/semi-structured) and who will be conducting the interviews.

Apologies for this omission. We have added that semi-structured interviews using a topic guide will be carried out by a member of the study team who was not involved in treatment (page 22, para 2).

Editor(s)' Comments to Author (if any):

*In the trial registry page you indicate that "The data sharing plans for the current study are unknown and will be made available at a later date". We are unable to consider a protocol manuscript on this basis, as we require a data sharing plan to be in place with the registry – as such, we require the authors to update the registry to indicate their data sharing plan before submitting a revised manuscript. The updated information should be consistent with what the authors state in the "Access to data" section in the manuscript.

The trial registry has been updated to include the information in the “Access to data” section in the manuscript.

*Please revise the article title format to enhance readability. Eg, “Therapist-supported online cognitive therapy for post-traumatic stress disorder (PTSD) in young people: protocol for an early-stage, parallel-group, randomised controlled study (OPTYC trial)” (or similar).

The title has been changed, thank you for your suggestion.

*Please clarify why you have decided to refer to the trial as “early stage” rather than “pilot” or “feasibility” (both in the title and throughout).

Our originally planned 3-arm trial (please see under Protocol Amendments, page 24) was referred to as a “feasibility” RCT. When we amended the design in response to the COVID pandemic (page 24), we decided to term the new 2-arm trial an “early stage” trial to distinguish it from the previous trial.

The term “early stage” is also used on the ISRCTN register and in our REC documentation. I hope it is now super clear from the abstract and full text that the primary study outcomes are data on feasibility and acceptability.

*We have amended the status of the consent/assent forms file to become a supplementary file to be included for publication, as per the SPIRIT checklist requirement. Please ensure that this supplementary file is cited in an appropriate place in the manuscript.

Thank you for doing this. I have added, “Please see supplementary files for copies of consent and assent forms” to the consent and assent section (page 24).

*Please revise the abstract and main text to ensure that you emphasise that the actual primary outcomes of the study are the feasibility/acceptability measures (as is appropriate for a pilot/feasibility study, and as is indicated in the trial registry page). The actual outcome measures for these feasibility/acceptability measures should be specified, as should the timepoint(s) at which they will be assessed.

Apologies – I can see that this was not emphasised sufficiently.

In the abstract, under method and analysis, I have added that the “primary study outcomes are data on feasibility and acceptability, including recruitment, adherence, retention, and adverse events” (page 2).

In the main text, under Outcomes (top of page 13), I have added, “The primary outcomes for the study are data on feasibility, adherence, and acceptability, which will be reported using the metrics specified below.”

*Please ensure that the manuscript is consistent with the trial registration page in terms of all outcomes defined – eg, we note that the registry lists nine secondary outcomes, whereas there seem to be fewer listed in the manuscript.

Thank you for pointing this out and my apologies for the inconsistency. I have updated the registry so that it is now consistent with the manuscript. I have not changed the manuscript, as it was correct.

*Please amend the “Methods and analysis” section of the abstract to include information on the study setting (ie, it will take place at how many centres, of what type, in what city/region/country). This information should also be clearly stated in the main text.

I have amended the method and analysis section of the abstract to include information on the study setting ie that recruitment will be “from 14 NHS Child and Adolescent Mental Health Services (CAMHS) in London and southeast England, from secondary schools and primary care in the same region, or via self-referral from anywhere in the UK using the study website”.

This information about study setting is also stated in the main text on page 9 para 2 under “Study Setting”

*Please update the 'Ethics and dissemination' section of the abstract to include a brief sentence on informed consent/assent.

I have updated the "Ethics and dissemination" section of the abstract on page 3 by including the following sentence: "For participants aged under 16, informed consent will be provided by carers and the young person will be asked for their assent; participants aged 16 years or older can provide informed consent without their parent or caregiver's involvement"

*Please amend the main text heading "METHODS: PARTICIPANTS, INTERVENTIONS, OUTCOMES" to "Methods and analysis"

This has been done (page 8).

and please move the "Trial design" subsection down to become the first subsection of the "Methods and analysis" section.

This has been done (page 8).

Please revise the subsequent subheadings to avoid the "METHODS:" format (as all subheadings within this section are part of the methods).

I have deleted "METHODS: ASSIGNMENT OF INTERVENTION" (from page 18), deleted "METHODS: DATA COLLECTION, MANAGEMENT, AND ANALYSIS" (from page 19), and deleted "METHODS: MONITORING" (from page 22).

*In an appropriate place within the main text 'Methods and analysis' section, please add some text to indicate the current study status and expected timeline for completion of the trial.

The date of the first and last recruited participant has been added to "Participant timeline" along with the current trial status on page 17.

*Currently you have a "Declaration of interests" section within the 'Ethics and dissemination' section of the main text – this should be moved to the end of the manuscript, replacing the current "Competing interests" statement which incorrectly states that there are "none".

I have moved this section down to replace the "competing interests" section (now on page 27).

Additionally, the statement should be updated to clarify which of the investigators (assuming this refers to named authors?) "provide training in the delivery of CT-PTSD, for which they may sometimes receive payment".

I have clarified that this refers to authors, and I have provided initials (page 27).